# The Internet of Things in the Nutritional Management of Patients with Chronic Neurological Cognitive Impairment: A Scoping Review

**DOI:** 10.3390/healthcare13010023

**Published:** 2024-12-25

**Authors:** Marco Sguanci, Stefano Mancin, Andrea Gazzelloni, Orejeta Diamanti, Gaetano Ferrara, Sara Morales Palomares, Mauro Parozzi, Fabio Petrelli, Giovanni Cangelosi

**Affiliations:** 1A.O. Polyclinic San Martino Hospital, 16132 Genova, Italy; sguancim@gmail.com; 2IRCCS Humanitas Research Hospital, via Manzoni 56, 20089 Rozzano, Italy; 3Bambino Gesù Children’s Hospital, Piazza Sant’Onofrio, 4, 00165 Rome, Italy; agazzelloni@gmail.com; 4Veneto Institute of Oncology IOV-IRCCS, 35128 Padua, Italy; orejeta.diamanti@iov.veneto.it; 5Department of Nephrology and Dialysis, Ramazzini Hospital, 41012 Carpi, Italy; amaranto1984@libero.it; 6Department of Pharmacy, Health and Nutritional Sciences (DFSSN), University of Calabria, 87036 Rende, Italy; sara.morales@unical.it; 7School of Nursing, University of Milan, “San Paolo” Campus, Asst Santi Paolo e Carlo, 20142 Milan, Italy; mauro.parozzi@asst-santipaolocarlo.it; 8School of Pharmacy, Experimental Center and Public Health “Stefania Scuri”, 62032 Camerino, Italy; fabio.petrelli@unicam.it; 9Units of Diabetology, ASUR Marche, 63900 Fermo, Italy; giovanni.cangelosi@virgilio.it

**Keywords:** Internet of Things, home healthcare, nutrition, sensors, chronic neurological cognitive impairment

## Abstract

Background/Objectives: The Internet of Things (IoT) technology connects objects to the internet, and its applications are increasingly used in healthcare to improve the quality of care. However, the use of IoT for the nutritional management of patients with chronic neurological cognitive impairment is still in development. This scoping review aims to describe the integration of IoT and its applications to support monitoring, interventions, and nutritional education for patients with chronic neurological cognitive impairment. Methods: A scoping review was conducted using the Cochrane, PubMed/Medline, CINAHL, Embase, Scopus, and Web of Science databases following the Arksey and O’Malley framework. Results: Of the 1424 records identified, 10 were included in the review. Most of the articles were peer-reviewed proceedings from technology conferences or publications in scientific and technology journals. IoT-based innovations in nutritional management were discussed in methodological articles, case studies, or project descriptions. Innovations were identified across three key areas: monitoring, intervention, and education. Conclusions: IoT technology offers promising innovations for the nutritional management of patients with chronic neurological cognitive impairment. However, IoT capabilities in this field are still in the early stages of development and are not yet highly specific.

## 1. Introduction

Chronic neurological diseases, such as Alzheimer’s disease, Parkinson’s disease, stroke, and dementia, are among the leading causes of disability and mortality worldwide; in 2021, Alzheimer’s disease and other forms of chronic neurological diseases ranked as the seventh leading cause of death, killing 1.8 million lives [1]. These diseases often lead to chronic neurological cognitive impairment, manifesting as difficulties in memory, attention, and executive functions, which significantly impact patients’ quality of life and daily functioning [2]

Chronic neurological cognitive impairment can significantly impact patients’ ability to maintain a balanced diet due to difficulties in meal planning, food preparation, and recognition of nutritional needs, highlighting the critical importance of personalized nutritional management to support overall health and cognitive function in this population [3]

Technological development has pushed the healthcare sector to explore innovative approaches to patient monitoring and care also in this particular condition [4,5,6].

One promising approach is the application of Internet of Things (IoT) technology, an intelligent network of interconnected physical devices equipped with sensors, software, and other technologies that enable them to collect, exchange, and act on data over the internet without requiring direct human interaction. This technology has the potential to revolutionize the monitoring and management of the health of patients affected by chronic neurological diseases, including their nutritional status [7,8,9]. People with moderate cognitive impairment often struggle with preparing and consuming appropriate foods [6], increasing their risk of malnutrition.

Studies have indicated that cognitive impairment is a common consequence of chronic neurological diseases like stroke, Alzheimer’s disease, Parkinson’s disease, and dementia, and can significantly impact long-term prognosis and quality of life.

Cognitive impairment following a stroke is associated with increased disability and dependence in activities of daily living [10]. Adequate nutritional monitoring in patients with Alzheimer’s disease, Parkinson’s disease, and dementia [11], and stroke [12] is essential for improving quality of life, as highlighted by recent studies. Cognitive impairment negatively affects the ability to perform independently the activities of daily living and complicates proper nutritional management in these patients.

The integration of IoT-enabled devices and sensors can provide the continuous, real-time monitoring of patient vital signs, activity levels, and dietary intake [8]. By collecting and transmitting these data to healthcare providers, IoT systems can facilitate the early detection of nutritional deficiencies or other health issues, allowing for timely interventions to prevent complications and improve patient outcomes. Moreover, IoT-powered wearables and mobile applications can empower patients with chronic neurological diseases like Parkinson’s disease [13] to take a more active role in managing their health, including monitoring their diet and physical activity [8,14].

The existing research has demonstrated the potential of IoT-based solutions in managing chronic conditions like diabetes, where the technology has been used to track patients’ diet, physical activity, and medication adherence. Similarly, IoT devices can be leveraged to enhance the quality of life for patients with chronic neurological diseases by providing personalized nutrition recommendations, reminders, and feedback to support their recovery and long-term well-being. A review of the literature on IoT in healthcare reveals several key insights: IoT-enabled devices can improve patient monitoring and care, though security and privacy remain critical concerns that must be addressed [4,8,9,10,12,14,15]. Ongoing research and development in the field of IoT in healthcare aims to tackle these challenges and unlock the full potential of this technology to transform the lives of patients with chronic neurological diseases.

The potential benefits of IoT in healthcare extend beyond stroke patients to include a broader range of chronic degenerative diseases. For individuals managing conditions such as diabetes, heart disease, post-stroke recovery, or age-related cognitive impairment, IoT-enabled devices offer a transformative approach to home nutrition monitoring and management. For instance, smart sensors in refrigerators or pantries can track food consumption patterns, while wearable devices can monitor blood sugar levels or other relevant physiological indicators. This continuous flow of data provides valuable insights into a patient’s eating habits and overall health [16]. By integrating this information with personalized nutritional recommendations and alerts, healthcare providers can better manage these patients by identifying potential warning signs and intervening proactively to prevent complications [17]. This personalized approach to nutrition management empowers patients to take control of their health, improves adherence to dietary guidelines, and ultimately enhances their overall well-being [18,19]. Similar strategies have proven effective in the management of chronic diseases, where mobile health (mHealth) technologies support self-care and contribute to improving patient outcomes. Some studies have explored the use of telemedicine interventions to promote physical activity and improve diet quality in post-stroke patients, showing promising results in terms of health and well-being [20,21]. Similar approaches could be adapted to optimize nutritional management and enhance overall health monitoring in patients with chronic neurological diseases.

The aim of this scoping review is to describe and map the use of the Internet of Things and its applications, either currently available or under development, in the management and monitoring of nutrition in patients with chronic neurological cognitive impairment to support the knowledge of healthcare personnel and help them in the management of this particular type of patients.

## 2. Methods

The review was reported according to the Preferred Reporting Items for Scoping Reviews (PRISMA-ScR, see Appendix A) [22]. For this scoping review, we adopted the methodology outlined by Arksey and O’Malley [23], which provides a structured framework for mapping a broad range of evidence, including emerging, quantitative, qualitative, or mixed studies. Preliminary registration in the OSF database was produced (https://doi.org/10.17605/OSF.IO/4ESDC). The primary aim is to identify key sources and highlight potential gaps in the current body of knowledge. Arksey and O’Malley’s approach begins with formulating the research question, followed by the identification and selection of relevant studies, data extraction, synthesis of findings, and, in some cases, consultation with stakeholders.

### 2.1. Identification of the Research Question

For the present study, the research question was formulated using the PCC model [24]. The PCC model is a framework used to refine research topics by focusing on three core elements: population (P), concept (C), and context (C). In this review, the following aspects were considered based on this approach: P: Patients with chronic neurological cognitive impairment (Alzheimer’s disease, Parkinson’s disease, stroke, and dementia). C: What is the role of IoT and related technological applications in managing and monitoring nutritional status? C: Where are (home, remote healthcare, and medical centers) these applications being implemented?

### 2.2. Identification of Studies Relevant to the Research Question

The search was carried out in August 2024 in the databases PubMed/Medline, Embase, CINAHL, Scopus, and Web of Science, using the following keywords: “Internet of Things”, “chronic neurological cognitive impairment”, and “nutrition” and their variations, opportunely combined by Boolean operators. A manual search was conducted scanning the reference lists of relevant articles and Google Scholar to retrieve additional records; full search algorithms are available in the data available statement (Appendix A).

### 2.3. Inclusion and Exclusion Criteria

#### 2.3.1. Inclusion Criteria

Primary and secondary studies: any type of experimental or observational study design, including qualitative and mixed-method studies;Related to the management and monitoring of nutritional status in chronic stroke patients;Participants: patients with chronic neurological cognitive impairment over 18 years;Secondary studies related to the topic of review;Proceedings of Congress contribution.

#### 2.3.2. Exclusion Criteria

Studies not related to the topic of review and that do not meet the inclusion criteriaBook chapters or editorials.

### 2.4. Data Charting Process

In accordance with the Arksey and O’Malley framework, a rigorous and systematic protocol was adopted for the “Charting of Information and Data” phase. This study strictly followed the standards established by the PRISMA-ScR Checklist [22], a widely recognized guideline specifically designed for scoping reviews. PRISMA-ScR provides clear and specific criteria, including both the essential reporting elements and optional additional components, tailored to address the unique challenges of scoping reviews.

The researchers (MS and SM) underwent a stringent process to select and include publications in the study. The authors individually reviewed all the titles and abstracts identified through the searches in the electronic databases. Duplicates and irrelevant records were removed using the Endnote 20 software (https://endnote.com/) [25].

Disputes were resolved with the involvement of a third reviewer (GC). The eligibility of full-text studies obtained was assessed by the two reviewers (MS and SM) independently based on the previously established criteria. Disagreements were resolved through a third reviewer (GC) (Figure 1). This methodological approach focused on identifying and abstracting key themes, interventions, primary findings, and other relevant information in alignment with the research objectives. All the information gathered from the different sources was subsequently consolidated into a unified framework, ensuring a consistent and comprehensive analysis that captures the essence and complexity of a scoping review.

### 2.5. Data Extraction and Synthesis

The following data were extracted: author, year, country, study design, sample size, comorbidities, setting, topic, Tele-healthcare intervention, IoT tool, Objective, or Findings. The included studies were categorized based on the identified review objectives and summarized through narrative synthesis.

## 3. Results

A total of 1424 articles were identified through the database searches: 176 from PubMed-Medline, 243 from Embase, 65 from CINAHL, 804 from Scopus, 100 from Web of Science, and 36 from other sources (IEEE Xplore, ACM Digital Library). After 297 duplicates were removed, all the titles and abstracts were screened, and the articles were evaluated for eligibility. Of these, 1102 were judged not to be relevant, and the remaining 25 full texts were assessed. A total of 15 of these were subsequently excluded as they did not meet the selection criteria for our research (7 for different population studies and 8 for different outcomes evaluated). The screening process ultimately included 10 studies in this scoping review (Figure 2).

### 3.1. General Characteristics of the Studies Included

Most of the studies were conducted in Western countries (n = 9; 90%) [26,27,28,29,30,31,32,33,34]. The articles featured a variety of study designs, including randomized controlled trials (RCTs) (n = 7; 77.8%) [24,26,27,29,31,33,34], pilot study (n = 1; 11.1%) [27], quantitative observational studies (n = 1; 11.1%) [30], and systematic review (n = 1; 11.1%) [32] (Table 1).

The studies were conducted in several countries, including the USA (n = 3; 33.3%) [26,33,34], Australia (n = 1; 10%) [35], Canada (n = 2; 22.3%) [27,29], the United Kingdom (n = 1; 10%) [28], Italy (n = 1; 10%) [30], Germany (n = 1; 10%) [31], and Portugal (n = 1; 10%) [32].

The combined patient sample from these studies totaled 710, ranging from as few as 9 to as many as 200 patients per study.

The included studies were divided by the topics of investigation: intervention, management/monitoring, and education. Most of the studies focused on monitoring and management (50%) [27,28,29,30,34]; only two studies investigated the educational aspect (20%) [26,31], and three reports described clinical interventions (30%) [30,31,33]. Table 1 summarizes the general characteristics of the included studies.

This scoping review aimed to explore the use of IoT and related technologies applications to improve nutritional outcomes and quality of life in individuals with previous stroke and cognitive impairment. The results were analyzed according to the purpose of using IoT technologies (Table 2 and Figure 3).

**Table 2 healthcare-13-00023-t002:** Data extraction from included studies.

Author Year	Country	Study Design	Sample	Comorbidities	Setting	Topic	Tele-Healthcare Intervention	IoT Tool	Objective or Findings
Eskin et al., 2012 [27]	Canada	Pilot Study	NA	Cognitive impairment Dementia	Home	Monitoring	Nutritional analysis monitoring dietary patterns	Image capture Food recognition algorithms based on machine learning Portion estimation tools	Achieved an 87.2% recognition accuracy Achieved reliable nutritional assessments by identifying and estimating food portions
Rimmer et al., 2013 [34]	USA	RCT	Patients (n = 9) *	Stroke	Home	Monitoring	Weight management program Physical activity plus nutrition (POWERS^plus^)	Telephone coaching Web-based remote coaching tool POWERS and POWERS^plus^ groups demonstrated a greater reduction in body weight compared with the control group	Significant group–time interaction in the post-intervention body weight POWERS and POWERS^plus^ groups demonstrated a greater reduction in body weight compared with the control group
Kosch T al., 2018 [33]	USA	RCT	Patients (n = 12)	Cognitive impairment	Home	Intervention	Food preparation	Smart intelligence kitchen	Creation of calorie- and nutrition-aware contextual cooking plans Semantic cookbook to share your recipes between smart kitchens and display them on an output device
Casaccia et al., 2019 [30]	Italy	Quantitative observational study	NA	Cognitive impairment	Senior center	Management	Self-manage (various activities including nutrition)	Modular integrated platform Open Application Programming Interfaces Wearable devices Lifestyle monitoring systems	NA
Amella et al., 2020 [26]	USA	RCT	Patients (n = 60)	Cognitive impairment Dementia	Respite Care Centers	Education	Trainer meal-time intervention	Video conferencing Digital communication platforms C3P Model	Weight maintenance or gain of the person with dementia Improving the QoL of caregiver
			Caregivers (n = 60)						
English et al., 2021 [35]	Australia	RCT	Patients (n = 80)	Stroke	Home	Intervention Education	Dietary Intervention (AusMed diet program) Provision of supporting resources Education about physical activity and healthy eating	“Attend Anywhere” system “Zoom” video conferencing	Feasibility Safety Reducing recurrent secondary stroke risk factors (blood pressure, physical activity levels, and diet quality)
									Fatigue, mood, and quality of life at 3, 6, and 12 months
Sakakibar et al., 2022 [29]	Canada	RCT	Patients (n = 126)	Stroke	Home	Management	Stroke Coach telehealth self-management program	SmartDiet Questionnaire	The post-intervention results showed no significant differences in lifestyle
Scheerbaum et al., 2022 [31]	Germany	RCT	Patients (n = 200)	Cognitive impairment	Home	Education	Nutritional counseling	CCT Online counseling	NA
Neves et al., 2022 [32]	Portugal	Systematic Review	NA	Cognitive impairment Dementia	NA	Intervention	Food intake detection	Inertial sensors Acoustic sensors, Cameras Electroglotography and piezoelectric sensors	Mapping the use of sensors for the detection of food intake episodes is an exciting field
Fletcher-Lloyd et al., 2023 [28]	UK	RCT	Households (n = 73)	Cognitive impairment Dementia Alzheimer’s Disease	Home	Monitoring	Monitoring living activities	PIR motion sensor (kitchen) Fridge door sensor Kettle sensor Oven sensor	Increase in day-time kitchen activity Significant decrease in night-time kitchen activity Potential changes in eating and drinking habits

Legend: RCT: randomized controlled trial; C3P: Change the Person, People, and Place; QoL: quality of life; PIR: passive infrared motion; CCT: computerized cognitive training; POWER: Personalized Online Weight and Exercise Response System; *: total participants of the study n = 102.

### 3.2. Education

Two studies [26,31] described the application of IoT technologies aimed at providing nutritional education for both patients and caregivers.

Amella et al. [26] demonstrated the effectiveness of a telemedicine-based meal training program through the use of video conferencing and digital communication platforms, which led to weight maintenance or gain for the person with dementia and improved the quality of life for the caregiver. The study analyzed an approach called “Partners at Meals” (PaM), which involved a “train-the-trainer” program for volunteers at Respite Care Centers (RCCs). The program included two initial training sessions (60–90 min) conducted by a research team trained in the C3P (Change the Person, People, and Place) method. The telemedicine platform used was “Doxy.me”. The PaM program is based on the C3P model, which focuses on three factors that influence eating behavior: the person with dementia, the people involved in the meal, and the place where the meal takes place. However, the study does not report preliminary results but lays the foundation for examining the effectiveness and sustainability of a telemedicine program to help families manage meals at home while promoting the quality of life for both the caregiver and the person with dementia.

Similarly, Scheerbaum et al. [31] described a fully digital study protocol for an RCT aimed at evaluating the effectiveness of computerized cognitive training (CCT) tools and online group nutritional counseling for people with mild cognitive impairment.

Individualized and basic computerized cognitive training tools are used, both delivered via software designed to improve the cognitive abilities of participants. The software uses machine learning algorithms to adjust the difficulty of the games based on the individual user’s performance. Online group nutritional counseling is also provided: the intervention involves biweekly 1.5 h online group sessions, focusing on a whole-food, plant-based diet or a healthy diet according to the guidelines of the German Nutrition Society.

The IoT technologies used in the protocol include video conferences, telephone interviews, and online questionnaires.

Studies demonstrate the potential of IoT technologies to provide scalable and personalized nutrition education for patients and healthcare providers. Strengths include the use of telemedicine platforms such as “Doxy.me” and adaptive tools such as computerized cognitive training (CCT), which provide remote, flexible, and personalized interventions. However, limitations include a lack of reported outcomes, potential accessibility barriers for underserved populations, and challenges in maintaining user engagement over time. Future efforts should focus on improving accessibility, ensuring robust evaluation, and improving long-term adherence to IoT-based programs.

### 3.3. Monitoring and Management

Monitoring and management were investigated in five studies (50%) [27,28,29,30,34].

Eskin et al. [27] described a “Smart Nutritional Assessment System” designed to help people with dementia monitor their diet effectively from home.

The technologies used are based on a “Computer Vision” system, for monitoring dietary intake, that uses food recognition and portion estimation algorithms to analyze images of meals. A photographic dataset of common foods is used to train the algorithms, with a webcam placed above the plate to capture the meal’s image. The system begins with plate segmentation and then focuses on food recognition. A method based on the color and shape of the plate is used, after which food recognition occurs. Different types of features, such as color, texture, and shape, are explored to describe the food in the image, and different classifiers (logistic regression, artificial neural networks, and support vector machines) are used to determine the food category. The information is stored in an image dataset to train and evaluate the algorithms. The system achieved an 87.2% accuracy in food recognition using the dataset created for the project, but it does not monitor meal frequency, eating habits, or dietary adherence.

Fletcher et al. [28] analyzed a Markov chain model to identify changes in the daily activity patterns of people living with dementia. This model is used to analyze kitchen activity data and identify stable behavior patterns, aiming to detect any anomalies, and emphasizing its role in detecting malnutrition and dehydration risks linked to behavioral changes. However, the model cannot assess specific nutritional intake or hydration levels directly, and it relies on indirect activity-based markers, which may not capture subtle nutritional issues or behavior deviations.

The sensors included a passive infrared motion sensor, a refrigerator door sensor, and smart plugs for appliances (kettle and oven).

The study presents three case studies that demonstrate how the Markov chain model can detect changes in behavior in people with dementia. Malnutrition and dehydration are strongly associated with cognitive and functional deterioration in people with dementia. The study highlights the importance of detecting changes in daily activity patterns through home monitoring via IoT technologies, which can monitor and quantify behavior patterns. The Markov chain model effectively identified behavior changes using home monitoring data.

The randomized controlled trial by Sakakibara et al. [29] described the use of a telemedicine coaching program called “Stroke Coach” to improve self-management for secondary prevention after stroke. The study collected data from 126 participants diagnosed with stroke. These patients were followed by a telemedicine coaching program that included individual teleconsultation sessions with a specialized coach. The study used the SmartDiet questionnaire to monitor the nutritional aspect, including daily fat consumption, medication adherence, and body composition; however, the direct, automated monitoring of dietary adherence or physical markers is not included.

The Stroke Coach system led to significant improvements in quality of life and glucose control (HbA1c) compared to the control group, suggesting its effectiveness in the nutritional management of patients post-stroke.

The study project by Casaccia et al. [30] presents a similar project, RESILIEN-T, which aims to develop a modular Information Communication Technology (ICT) system to help people with early cognitive impairment self-manage their health. The project involves wearable sensors (smartwatches), home monitoring systems, and tablets for communication and data management. While specific results were not presented, the system architecture and technological approaches were described. The system offers three versions—Basic, Plus, and Home—focused on self-management of health. The Basic version includes basic nutrition services, providing users with content and nutritional suggestions via an app. The monitoring capabilities are limited: for example, it lacks the integration of real-time dietary intake analysis or behavioral tracking that could enhance its application in nutritional management.

Rimmer et al. [31] examined the effect of a 9-month, telephone-based remote weight monitoring and management program for people with physical disabilities using a web-based system called “Personalized Online Weight and Exercise Response System” (POWERS).

With a total sample of one hundred two participants with physical disabilities, the study included nine stroke patients; they were enrolled and randomized into one of three conditions: exercise alone (POWERS) (n = 4), exercise plus nutrition (POWERSplus) (n = 2), and control (n = 3). The POWERSplus group received an identical intervention to the POWERS group but with added nutritional support. Post-intervention differences in body weight were found between groups. The POWERSplus group demonstrated greater weight reduction compared to the control group.

However, monitoring nutritional adherence and exercise compliance is largely participant-driven, using self-reports and periodic interventions, which may reduce accuracy and introduce variability. Long-term sustainability and the monitoring of other factors like psychological or social determinants of dietary habits are not addressed.

The study suggests that a low-cost telephone intervention supported by a web-based remote coaching tool may be an effective strategy for helping overweight adults with physical disabilities maintain or reduce their body weight while adhering to an adequate nutritional regimen.

Overall, the studies demonstrate significant advancements in monitoring dietary intake, activity patterns, and weight management, but challenges remain in areas like comprehensive real-time dietary adherence tracking, hydration monitoring, and the integration of psychological or environmental factors affecting nutrition.

### 3.4. Intervention

English et al. [35] presented the protocol for a pilot randomized controlled trial to evaluate the feasibility, safety, and potential efficacy of a 6-month physical activity and/or diet (PA and/or DIET) intervention delivered via telemedicine for people who have had a stroke. The project will enroll 80 adults who have experienced a stroke, live at home with internet access, and can perform physical activity. The trial will deliver PA and/or DIET interventions via telemedicine, using video calls to provide support and instruction. Parameters will be monitored using an activPAL device (PAL Technologies Ltd., Glasgow, Scotland, UK, for physical activity), a blood pressure monitor, and a glucose monitor. Nutrition will be managed through telemedicine support, including education, resources, and advice from an accredited dietitian on adherence to the AusMed (Mediterranean-style) diet. Although the results are not yet published, the study emphasizes the importance of physical activity and a quality diet in secondary stroke prevention.

The recent systematic review by Neves et al. [32] focused on the various methods and technologies used for food intake detection. The article analyzes 30 studies that employed different technologies, such as cameras, inertial sensors, acoustic sensors, electrogastrography, and piezoelectric sensors, to detect food intake in patients with conditions like diabetes, functional dyspepsia, and cognitive impairment. The number of participants varied from study to study, ranging from a minimum of 6 to a maximum of 100. The review highlights that there is still no standard method for food intake detection, though methods based on deep learning and neural networks are the most commonly used.

Cameras are the most frequently used sensors for food recognition and classification, but the system still requires user participation to take pictures of the food. Acoustic sensors present challenges in terms of positioning and sensitivity. While integrating multiple sensors could improve the accuracy of food intake detection, further research is needed to enhance accuracy, feasibility, and acceptability for people with cognitive impairment.

Another RCT by Kosh et al. [33] explored the design requirements for smart kitchens tailored to people with cognitive disabilities. The study involved 12 people with cognitive disabilities, along with four employees and volunteers who supervise residents, participating in interviews. The participants were between 30 and 49 years old. The study highlights the potential of using interactive technologies to support common kitchen activities in assisted living facilities. These technologies could reduce the workload of volunteers and allow residents to learn cooking skills more independently, thereby improving their nutritional status.

Lastly, the studies highlight a variety of technological interventions aimed at improving health monitoring and management. These include telemedicine-based programs, such as the “Stroke Coach” and the protocol by English et al. [35], which combine virtual consultations with wearable devices to monitor physical activity, blood pressure, and glucose levels, providing personalized support for dietary and exercise management. Other interventions focus on detecting food intake through advanced technologies, such as acoustic, piezoelectric, and visual sensors, often supported by deep learning algorithms for food recognition. Finally, interactive technologies like smart kitchens aim to promote autonomy in daily activities, helping individuals with cognitive disabilities improve meal preparation and, consequently, their nutritional status. These interventions reflect integrated approaches that leverage sensors, artificial intelligence, and interactive interfaces to address diverse clinical and behavioral needs.

However, these techniques show limitations, including the need for user participation, challenges in standardizing methods, and accessibility barriers for individuals with cognitive disabilities or limited digital resources.

## 4. Discussion

This scoping review examined the current technologies used to manage and care for the nutritional needs of individuals with chronic neurological cognitive impairment. The use of technology encompassed three domains: (1) monitoring and management, (2) intervention, and (3) education. Nutritional monitoring and assessment are essential components of the nutritional process, allowing for the detection and treatment of poor nutritional status, as well as understanding the effectiveness of interventions [36].

The most prominent area of technology use was software applications (50% of the studies) [27,28,29,30,31] designed to monitor and manage nutritional status. Casaccia et al. [30] explored the use of ICT technologies through wearable sensors (smartwatches), communication and data management systems, and home monitoring devices. Similarly, Fletcher et al. [28] and Sakakibara et al. [29] emphasized the key role of telemedicine in managing the nutritional aspects of individuals with cognitive impairment.

Telemedicine has certainly emerged as one of the most versatile and well-researched tools in the last decade, particularly for remote hospital management. A recent satisfaction survey revealed a high rate of satisfaction, with over 95% of the patients reporting positive experiences [37].

In populations with chronic neurological cognitive impairment, the specificity of nutritional needs requires personalized nutritional management. The greatest challenge is to keep the dietary plan consistently accessible to the user while monitoring and educating them through specific interfaces and programs, often supported by IoT technologies [38]. These tools provide personalized nutritional assistance by qualified healthcare professionals [39].

Cutting-edge approaches for automated food intake recording utilize image recognition to assess food and portion sizes, often using phone cameras [40,41]. Data are typically collected in cloud databases, where they are processed and stored, as demonstrated in a study by Eskin et al. [27], which aimed to help individuals with dementia monitor their diet effectively from home. This approach is supported by a systematic review by Neves et al. [32] and recent studies on the topic [42,43]. These systems enable meal plans to be accessible anywhere and act as a data collection mechanism for recording macronutrients, micronutrients, and hydration levels [38].

The reviewed applications recognize the specific needs of patients and the importance of personalized recommendations to improve nutritional outcomes. However, there is an increasing need for standardized application interfaces to improve usability and personalize content [44].

Nutrition education plays a key role for post-stroke patients. The most commonly used IoT tools for education are video conferencing and digital communication platforms. These systems employ various learning media, including apps, web-based platforms, game-based learning, and online tools [45]. Amella K. et al. [26] demonstrated the effectiveness of a telemedicine-based nutritional training program, which led to weight maintenance or gain in individuals with dementia and improved quality of life for caregivers. This was achieved through online counseling, as also described by Schherbaum et al. [26], who developed a program of CCT and online group nutritional counseling for individuals with mild cognitive impairment.

The included studies reflect a growing interest in the use of the IoT, such as fitness trackers and sensors, to support post-stroke patients. IoT technology enables the continuous collection of health data and the reporting of results in general chronic care and social impact [46,47,48,49,50]. This review highlights applications focused on the nutritional aspect, demonstrating that the integration of nutrition and technology can effectively support patients with cognitive impairment.

The development of such projects requires close collaboration between different professionals; nurses, doctors, and engineers collaborate to create shared knowledge and it is very important to manage collaborations according to precise protocols, as described in a recent study [47]. In the development phases, software system standardization is an essential first step, especially for monitoring, nutritional planning, and education, through simple and accessible platforms.

### 4.1. Challenges of Implementing IoT Technologies and Cost-Effectiveness Considerations

The implementation of IoT technologies in healthcare settings, particularly in the management of chronic diseases, presents several challenges that must be carefully considered. These difficulties are not unique to IoT but are also common in other healthcare interventions, such as those in cardiovascular health. Understanding these obstacles is crucial for effectively integrating IoT technologies into healthcare systems. For example, the South American Centre of Excellence in Cardiovascular Health (CESCAS) has highlighted ongoing studies that identify challenges in adopting complex interventions, such as the training of healthcare personnel, resistance to change, and the lack of adequate resources [51]. Moreover, the use of innovative technologies, such as virtual reality (VR) for training, has shown promising results in improving treatment effectiveness. However, similar challenges related to training and user acceptance have emerged in this context [52]. Although these solutions may involve high initial costs, the long-term benefits, such as improvements in treatment quality and the reduction in complications, suggest that the adoption of IoT technologies could be cost-effective in the long run [51,52]. In addition, practical challenges in the implementation of IoT technologies in healthcare settings include overcoming barriers such as the limited infrastructure in certain regions, which may hinder the widespread adoption of these solutions. Furthermore, the need for ongoing training and support to ensure the effective use of IoT systems remains a significant consideration. Despite these challenges, the potential of IoT to improve patient outcomes and streamline healthcare delivery makes its integration a worthwhile investment over time.

### 4.2. Strengths and Limitations

It should be noted that while this review provides an overview of the current applications of the Internet of Things in the management, monitoring, and nutritional education of patients with chronic neurological cognitive impairment, it has some limitations. Variability in study designs, heterogeneous patient samples, and differences in outcomes make it difficult to establish definitive recommendations. Additionally, the lack of standardization in the use of IoT technologies and the varied experimental approaches may impact the interpretation of the data.

One of the included studies was not specifically dedicated to patients with cognitive impairment, although a subsample of it included patients with stroke, and for this reason, it was not possible to extrapolate the results related to this specific sample.

Looking ahead, future research should focus on developing standardized IoT frameworks and tools that can be widely applied to nutrition management, especially in specialized settings to ensure the consistency and comparability of results. The integration of AI with IoT technologies could open opportunities for real-time adaptive interventions, tailored to individual patient needs, potentially improving the accuracy and effectiveness of monitoring and management. Furthermore, addressing data privacy and security challenges is crucial to building trust among users and ensuring widespread adoption. Another key challenge is bridging the digital divide by making IoT solutions accessible and affordable to all, especially in resource-limited settings. Finally, interdisciplinary collaborations between healthcare professionals, technologists, and policy makers will be essential to designing and implementing scalable and patient-centric solutions. The review is lacking in the section related to the perspective of the practitioners, which certainly represents a potential future avenue for primary and secondary research that could serve as a reference for the scientific community.

This review is intended as a starting point for future research and trials, which should aim to address these limitations through the adoption of standardized IoT tools and a focused approach to populations with these specific diagnoses. Possible systematic reviews could further validate the results obtained, although the review conducted maintains a certain level of reliability and scientific rigor on the topic addressed.

## 5. Conclusions

The nutritional management of patients with chronic neurological cognitive impairment, when performed remotely through the use of IoT technologies, remains a significant health challenge and is still underdeveloped. Nutritional monitoring, therapeutic interventions, and education require a highly personalized approach, involving targeted education, standardized treatment protocols, and a deep understanding of the interactions between technology and users. Our findings illustrate how the Internet of Things is transforming nutritional counseling and remote education processes: the use of this technology shows promising applications in the management of the risk of malnutrition in complex patients with cognitive impairment.

Currently, there is no standardized approach, as evidenced by the heterogeneity of the results. This indicates the need for further research and experimental designs with more rigor and on particular aspects to consolidate current knowledge and provide a solid foundation for future clinical interventions. The integration of IoT-based strategies into the existing care models could represent a significant step forward in the care of these particular pathologies.

## Figures and Tables

**Figure 1 healthcare-13-00023-f001:**
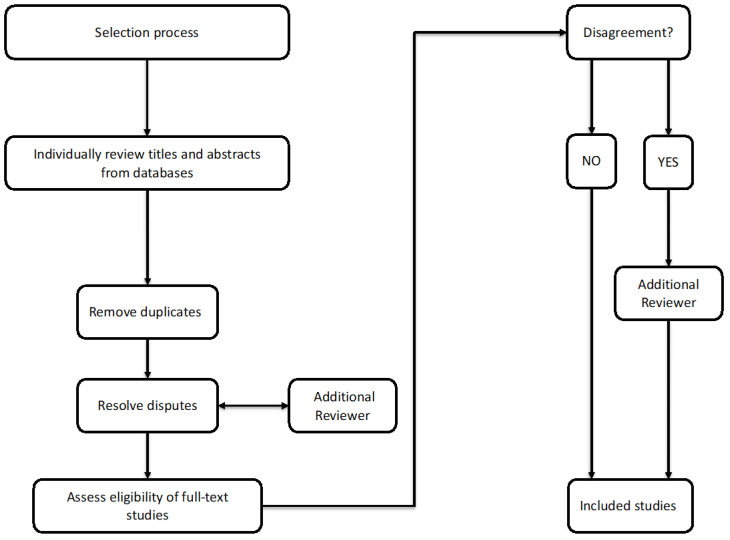
Data charting process flowchart.

**Figure 2 healthcare-13-00023-f002:**
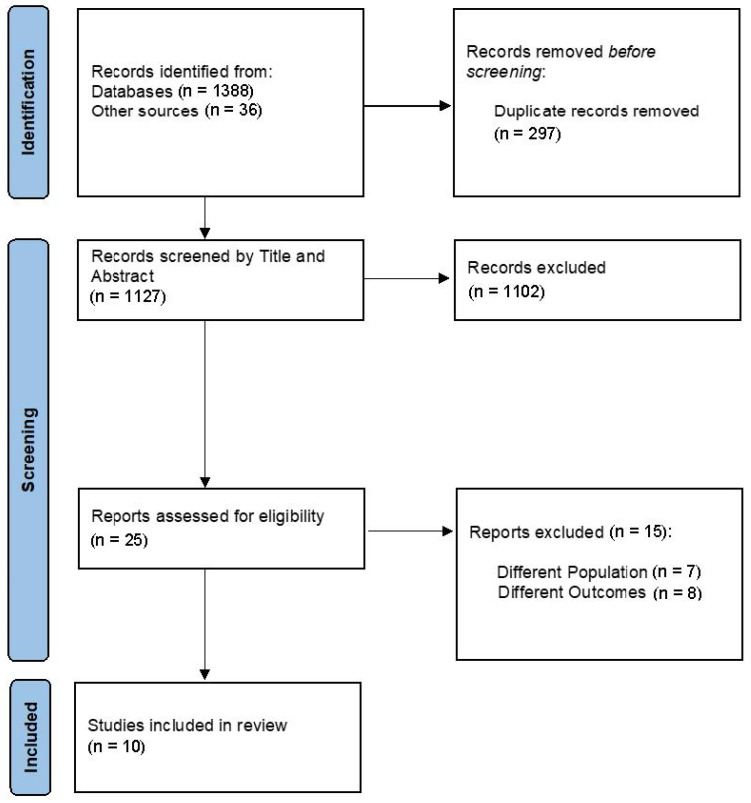
Prisma ScR flowchart.

**Figure 3 healthcare-13-00023-f003:**
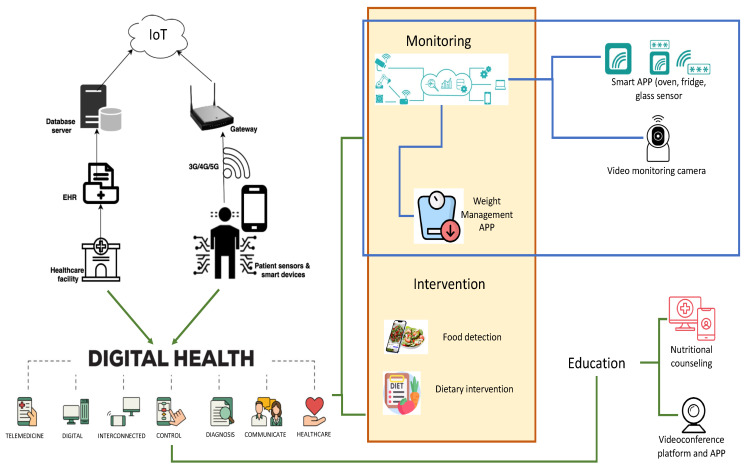
IoT application in home nutrition management. Legend: EHR: Electronic Health Record; IoT: Internet of Things; APP: Web or Mobile Application; 3G/4G/5G: Mobile Network Connectivity; green connectors: between Digital Health and application settings; blue connectors: internal correlations in application setting.

**Table 1 healthcare-13-00023-t001:** Characteristics of the studies included.

Characteristic	Frequency (n = 10)	Percentage
Publication year		
2023	1	10%
2022	3	30%
2021	1	10%
2020	1	10%
2019	1	10%
2018	1	10%
2013	1	10%
2012	1	10%
Geographical distribution		
USA	3	30%
United Kingdom	1	10%
Germany	1	10%
Canada	2	20%
Italy	1	10%
Portugal	1	10%
Australia	1	10%
Type of studies		
Pilot Study	1	10%
Quantitative Observational Study	1	10%
Randomized Control Study	7	70%
Systematic Review	1	10%

## Data Availability

Data sharing is not applicable. No new data were created or analyzed in this study.

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
