# Peer review of "The Internet of Things in the Nutritional Management of Patients with Chronic Neurological Cognitive Impairment: A Scoping Review"

_healthcare, 2024, doi:10.3390/healthcare13010023_

Round 1

Reviewer 1 Report

Comments and Suggestions for Authors

Paper presentation is not good. It should be systematic . 

Abstract should be clear and crisp. Vast review has been done but extraction of meaningful insights related to title of the paper is missing. 

Tables and figures are not explained.

The detail review comments are attached in the attached file. 

Comments on the Quality of English Language

English is average. Multiple use of 'or' is not recommended. Sentences should be complete and clear.

Author Response

Thank you for the comments. The replies to the comments are in the attached file.

Best Regards

The Authors

Reviewer 2 Report

Comments and Suggestions for Authors

- Abstract and conclusion require re-editing.

- Authors must mention that this sudy is to attended to help healthcare staffs.

- The work in this field is significant and useful, but the satisfaction of doctors and healthcare supervisors must be achieved.

- This work idea deals with the person's health and life, so one must be careful in proposing some conclusions and recommendations.

Author Response

(The authors gave the same response as above.)

Reviewer 3 Report

Comments and Suggestions for Authors

The paper presents a scoping review of using IoT for management of nutrition among post stroke patients. The paper has categorized the work in literature into several categories and provided a brief overview of all of them.

Comments:

1.      A comparison of the survey with other surveys available can be added. For example, are there any scoping surveys in this area. What about other types of survey papers?

2.      The strength and weakness of reviewed techniques can be better highlighted. Moreover, current challenges in each category in Section 3 can be presented.

3.      Section 3.3 can better highlight what sort of monitoring and management is achieved and what problems cannot still not be monitored.

4.      Section 3.4 can explain the type of interventions that are possible.

5.      More elaboration related to data collection by IoT, and its analysis can be added.

6.      There are very limited future opportunities and challenges in the paper. More elaboration is needed.

Author Response

(The authors gave the same response as above.)

Reviewer 4 Report

Comments and Suggestions for Authors

This manuscript, “Challenges and Opportunities in Implementing the Internet of 2 Things in Nutritional Management of Post-Stroke Patients 3 with Cognitive Impairment: A Scoping Review,” reports a scoping review of the use of the internet of things for nutritional management. Although the internet of things is a ‘trendy’ topic currently, I have several major concerns about this review that are detailed below.

Major:

1. The target population is unclear because it changes throughout the manuscript. For example, sometimes it is stated that stroke or cognitive impairment is the target (abstract and intro, lines 80-82); other times it is individuals with stroke who have cognitive impairments (title and results line 174); yet other times it is just patients with chronic stroke (methods, line 98). If that target was individuals with stroke and individuals with cognitive impairment, please justify why these two groups are lumped together as there are some very important differences between them.

2. The introduction should be revised based on the response to comment 1. For example, when I thought it was about individuals with stroke I needed more information about current status of nutrition after stroke. Depending on target population, more tailored information is needed to establish a clear rationale for this line of inquiry.

3. Along the lines of comment 1, when looking at Table 2, I am further confused by the target population as there appears to be studies that include multiple sclerosis and cerebral palsy.

4. Internet of Things is never defined. I recommend defining it in the introduction. After doing this, the authors will likely need to clean up some of the language throughout the manuscript where there are numerous qualifiers on IoT. Two examples include “role of IoT and related technological application” (methods, line 99) and “This scoping review aimed to explore the use of telemedicine interventions and IoT 172 technologies” (results, line 172-173).

Minor:

1. The first sentence is odd. Stroke did not “prompt” exploring new approaches for monitoring, advancements in technology did.

2. In the results section, the study design categories are unusual. Please use more standard and descriptive terms (i.e., what kind of qualitative study? What is a project study?) (line 157 and Table 1, 2).

3. The “geographical distribution” and “type of studies” sections of Table 1 are confusing because the percentages do not add up to 100%. I suggest removing the western/eastern countries and the primary/review categories to improve the clarity (that info can go in the text). Also check your percentages. For example, 2 out of 10 studies does not equal 22.3% (number of studies from Canada for example).

4. Figure 2 is not described enough. For example, what are the different color arrows? Also the font is too small (even at 200% zoom) to read a lot of the text, especially in the digital health section.

Author Response

(The authors gave the same response as above.)

Reviewer 5 Report

Comments and Suggestions for Authors

The authors present a scoping review of IoT applications in nutritional management for post-stroke patients and those with cognitive impairment. The review addresses an important and timely topic in healthcare technology.

1. Cosider expanding on quality assessment of included studies.

2. Include more discussions on implementation challenges of IoT technologies, also include cost-effectiveness considerations.

3. Elaborate on practical implications for healthcare providers and challenges faced in real-world settings.

Author Response

(The authors gave the same response as above.)

Round 2

Reviewer 3 Report

Comments and Suggestions for Authors

Authors have addressed comments of the last round.

Author Response

Dear Peer,

thank you very much for your comment and attention.

Best.

the Authors

Reviewer 4 Report

Comments and Suggestions for Authors

The target population is still concerning. In the title it says “Post-Stroke Patients with Cognitive Impairment” this implies that you are only focusing on individuals with stroke who also have cognitive impairments. Similarly in the Methods section 2.1 the population is identified as individuals with chronic stroke, including those with cognitive impairments. Again this suggests that you are focusing on individuals with stroke who have cognitive impairments. Based on the authors’ response and the table this is not true. Instead, you are focusing on people with stroke and people with cognitive impairment. I do not understand why you would include both of these groups. Specifically, if you are interested in people with cognitive impairment this could include individuals with stroke, dementia, etc, but why is the introduction so focused on stroke rather than on cognitive impairment? Additionally, per the authors responses they have included stroke and individuals with cognitive impairments because they have overlapping challenges in managing nutrition. Although this may be true, not everyone with stroke has cognitive deficits and, conversely, individuals with stroke have a number of other challenges including deficits in motor function that likely impact nutrition management.

I am concerned about this because it significantly impacts your search results. For example, Rimmer et al  (2013) focuses specifically on individuals with PHYSICAL disabilities. Yes, there are individuals with stroke included in their study, but neither stroke nor cognitive deficit was their focus. This is just one example of why I think the lack of clarity and preciseness in defining your population and questions is a significant problem. I think there is a need to refine your question and your target population. In other words, are you interested in stroke or in cognitive deficits? If you are interested in cognitive deficits, stroke could be a cause of that, but you would need to confirm that the sample has cognitive deficits.

Author Response

Dear Peer,

thank you very much for your effort and the answer in the annex file.

Best.

The Authors
